# Microvascular Dysfunction across the Spectrum of Heart Failure Pathology: Pathophysiology, Clinical Features and Therapeutic Implications

**DOI:** 10.3390/ijms25147628

**Published:** 2024-07-11

**Authors:** Giulia La Vecchia, Isabella Fumarulo, Andrea Caffè, Mario Chiatto, Rocco A. Montone, Nadia Aspromonte

**Affiliations:** 1Department of Cardiovascular and Pulmonary Sciences, Catholic University of the Sacred Heart, 00168 Rome, Italy; giulia.lavecchia22@gmail.com (G.L.V.); isabella.fumarulo01@icatt.it (I.F.); andreacaffe97@gmail.com (A.C.); 2Center of Excellence in Cardiovascular Sciences, Isola Tiberina Hospital Gemelli Isola, 00186 Rome, Italy; 3Department of Cardiovascular Sciences, Fondazione Policlinico Universitario A. Gemelli IRCCS, 00168 Rome, Italy; roccoantonio.montone@unicatt.it; 4Azienda Ospedaliera “SS. Annunziata”, 87100 Cosenza, Italy; mariochiatto53@gmail.com

**Keywords:** microvascular dysfunction, heart failure, chronic inflammation

## Abstract

Coronary microvascular dysfunction (CMD) plays a crucial role across the spectrum of heart failure (HF) pathology, contributing to disease development, progression, and outcomes. The pathophysiological mechanisms linking CMD to HF are complex and still not completely understood and include chronic inflammation, oxidative stress, and neurohormonal activation. Despite the diagnostic and prognostic relevance in patients with HF, there is no specific therapeutic strategy targeting CMD to date. Moreover, the diagnosis of this clinical condition is challenging. In this review article, we aim to discuss the different clinical pathogenetic mechanisms linking CMD to HF across the different spectra of these diseases, their prognostic relevance, and the possible therapeutic targets along with the remaining knowledge gaps in the field.

## 1. Introduction

Heart failure (HF) represents a complex and multifaceted clinical syndrome with a substantial global health burden worldwide [1,2]. An estimated 64.3 million people are currently dealing with heart failure [1]. In Western countries, the prevalence of this condition is around 1% and 2% but is estimated to continuously increase in the next few decades, especially because of an increasing and aging population. Coronary artery disease (CAD) is still the leading cause underlying HF in Western countries [1]. However, while more attention has traditionally been devoted to epicardial CAD, emerging research underscores the pivotal role of coronary microvascular dysfunction (CMD) in the pathogenesis and evolution of this clinical syndrome, especially heart failure with preserved ejection fraction (HFpEF) [3,4,5,6,7,8]. CMD refers to a spectrum of changes in the structure and function of coronary microcirculation, causing reduced coronary blood flow (CBF) and ultimately resulting in myocardial ischemia, fibrosis and remodeling [3,7,9]. Despite the emerging pathophysiological and prognostic role of CMD, there is no specific treatment to date addressing this condition. Moreover, the invasive and non-invasive diagnosis of CMD is still challenging. This narrative review aims to explore the molecular mechanisms underlying CMD, and its distinct manifestations across the spectrum of HF pathology, discussing novel therapeutic targets and remaining knowledge gaps in the field.

## 2. Pathophysiology of CMD

The coronary tree is composed of different interacting compartments with decreasing size and distinct capacitance and resistance functions. The proximal territory is composed of the epicardial coronary vessels, which offer a predominant capacitance function and little resistance to CBF [3]. Going more distally, the intermediate compartment includes the pre-arteriolar vessels (500 to 100 µm diameter), which are more sensitive to pressure and flow changes, displaying significant resistance to blood flow. Arterioles are the main contributors to the metabolic regulation of CBF by being more responsive to changes in the intramyocardial concentration of metabolites with direct vasoactive properties on the coronary endothelium [3]. Coronary microvascular dysfunction encompasses a combination of different structural and functional abnormalities at the microcirculatory level, involving pre-arterioles, arterioles, and capillaries [3,7,8,9,10,11]. Functional mechanisms may be due to impaired dilation and/or an increased constriction of coronary microcirculation (microvascular spasm) [12]. Reduced vasodilation can be due to mechanisms that are either dependent on or independent of the endothelium [3,7,13]. Endothelial-dependent mechanisms include the reduced production and/or enhanced degradation of endothelial-derived relaxation molecules, such as prostaglandins, nitric oxide (NO) and endothelium-derived hyperpolarizing factor(s) (EDHFs), as well as the enhanced release of vasoconstrictor mediators, such as endothelin-1 (ET-1) [3,7]. In this regard, chronic inflammation and free radical overproduction are considered key pathogenetic mechanisms driving the reduced bioavailability of coronary vasodilators. Preclinical studies showed that an increased concentration of intracellular reactive oxygen species (ROS) promotes the transformation of NO into free radicals by switching the NO synthetase from a NO to a ROS-producing enzyme, thus resulting in reduced vasodilation mediated by nitric oxide (NO) and the increased vasoconstriction activity of ET-1 through the activation of the RhoA/Rho-kinase pathway [14,15]. Endothelial-independent mechanisms are still not well known, but impaired vasodilator properties and enhanced susceptibility to normal vasoconstrictor stimuli of the vascular smooth muscle cells (VSMCs) at the microcirculatory level, as well as abnormal autonomic activity, are thought to be involved in this condition [7]. Beyond functional alterations, CMD may be due to structural abnormalities of the coronary microcirculation. Hence, vascular “remodeling” at the microcirculatory levels with hypertrophic inward remodeling (mainly due to smooth muscle hypertrophy and increased collagen deposition) and luminal narrowing of the arterioles and capillaries lead to pathological changes such as perivascular fibrosis, microvascular rarefaction, and myocardial stiffness that contribute to altered coronary physiology and reduced CBF [3]. These phenomena are extensively documented in clinical conditions characterized by an increased myocardial mass, such as in hypertensive heart disease and hypertrophic cardiomyopathy, where these pathological abnormalities extend beyond the vascular level toward the left ventricle, contributing to chronic myocardial ischemia, interstitial fibrosis, and progression to HF and increasing the risk of adverse cardiovascular outcomes [3,16,17] (Figure 1).

## 3. CMD across the Spectrum of Heart Failure Pathology

HF is a clinical syndrome characterized by symptoms and signs arising from impaired cardiac function. The cardiac dysfunction may be due to either structural or functional abnormalities leading to elevated pressures within the heart chambers and/or insufficient blood output at rest or during exertion, which are responsible for the clinical symptoms and raised biomarkers of cardiac dysfunction [2,18,19,20]. The underlying causes of HF can vary. In developed countries, CAD and hypertension are the most frequent, followed by valvular heart disease, arrhythmias, cardiomyopathies, systemic disease (i.e., neuromuscular, autoimmune, endocrine), and therapy-related issues (i.e., chemotherapy, radiotherapy) [21]. 

CMD may be present across the entire spectrum of HF, from HF with reduced ejection fraction (HFrEF) defined by a left ventricular ejection fraction (LVEF) < 40%, to HF with mildly reduced ejection fraction (HFmrEF) with an LVEF between 40% and 50%, to HF with preserved ejection fraction (HFpEF) with an LVEF > 50% [22,23] (Table 1).

Nevertheless, the pathophysiological role of CMD seems to be essential, especially in the latter group of patients, those affected by HFpEF, who account for half of all heart failure presentations [22]. HFpEF usually affects patients who are typically older and more often female compared to those with HFrEF and HFmrEF. Also, they usually suffer from several comorbidities, both cardiovascular, such as atrial fibrillation (AF), hypertension, and stroke, and non-cardiovascular, such as diabetes mellitus (DM), obesity, and chronic kidney disease (CDK) [22]. Diastolic dysfunction is the key clinical feature of HFpEF, a multifaced process due to chronic inflammation, cardiometabolic dysfunction, and extracellular fibrosis [22]. More recently, HFpEF has been redefined as a systemic disease that affects more than just the heart. It is now recognized to involve multiple organs, with systemic inflammation, extracellular fibrosis, and microvascular dysfunction [8,9,24]. In this regard, comorbidities such as hypertension, diabetes mellitus, obesity, and kidney disease play a pivotal role in contributing to the pro-inflammatory milieu underlying HFpEF and are independent risk factors for this clinical condition [22]. Over the past few years, a significant association between HFpEF and CMD has been found and CMD-HFpEF is turning out to be a prevalent endotype of HFpEF that seems to be associated with a poor cardiovascular prognosis [24,25].

The PROMIS-HFpEF (prevalence and correlates of coronary microvascular dysfunction in heart failure with preserved ejection fraction) study evaluated the prevalence and the prognostic significance of CMD, defined as a coronary flow reserve (CFR) < 2.5 measured non-invasively with adenosine stress transthoracic Doppler echocardiography, among 202 patients with a diagnosis of HFpEF [24]. The authors proved that nearly 75% of all patients with HFpEF showed evidence of CMD in the absence of significant epicardial CAD [24]. Moreover, worse CFR values were seen to be associated with risk markers of HF severity, such as higher N-terminal pro–B-type natriuretic peptide [NT-proBNP] levels and evidence of RV dysfunction (assessed by tricuspid annular plane systolic excursion (TAPSE) and right ventricular free wall strain) (*p* < 0.05) [24]. Recently, a meta-analysis by Lin at al., including results derived from 10 studies and involving 1267 patients, documented a prevalence of CMD around 71% in patients with HFpEF [25]. Again, Yang et al. documented a similar prevalence of CMD among HFpEF patients, with nearly the same proportion of endothelium-dependent and -independent CMD [26]. These results have been strengthened by the following studies evaluating the association between CMD and HFpEF by using either invasive assessment (e.g., coronary angiography) and non-invasive testing (e.g., cardiac magnetic resonance [CMR] with stress perfusion imaging and coronary vasodilator agents administration) [4,27]. Paolisso et al. investigated the presence of CMD, assessed invasively by continuous intracoronary thermodilution, in 56 patients with de novo HF and nonobstructive CAD [28]. According to the study results, 29 individuals (52% of the study population) were affected by CMD, with a similar prevalence across the spectrum of HFrEF and HFpEF pathology. However, different hemodynamic properties and phenotype characteristics were found in patients with HFrEF compared to those with HFpEF. Indeed, in the HFrEF group, CMD was described as a “functional” alteration, characterized by lower absolute microvascular resistance and higher absolute coronary flow values at rest. Conversely, in the HFpEF group, CMD was classified as a “structural” pathology, characterized by higher absolute microvascular resistance and lower absolute coronary flow during hyperemia [28]. These findings may suggest that the correlation between CMD and HFpEF has to be sought in diastolic dysfunction and its interdependence with capillary rarefaction, which is actually “structural” CMD. Strengthening this hypothesis, an autopsy study conducted by Mohammed et al. reported the presence of a significant capillary rarefaction in patients affected by HFpEF, which also correlated with the presence and extension of myocardial fibrosis [5]. The patients with HFpEF also had more cardiac hypertrophy and epicardial CAD compared to the controls, which may contribute to the left ventricular diastolic dysfunction typical of HFpEF [5]. Capillary rarefaction has also been documented using non-invasive techniques. Arnold et al. reported lower myocardial perfusion reserve (MPR) values assessed by CMR in HFpEF patients compared to healthy controls [27]. On the other hand, the authors did not find a significant correlation between reduced MPR, suggestive of CMD and capillary rarefaction, and myocardial fibrosis.

Despite the clinically and epidemiologically significant association between CMD and HF, the European Society of Cardiology (ESC) guidelines do not recommend a routinary assessment of CMD in HF patients and only provide a class IIa recommendation for invasive testing (guidewire-based CFR and/or microcirculatory resistance measurements) and a class IIb recommendation for non-invasive testing (transthoracic Doppler of the left anterior descending (LAD), CMR, and positron emission tomography (PET)) in the context of chronic coronary syndromes (CCSs) [3,34]. However, invasive testing of CMD is increasingly adopted in clinical practice, and its use is expected to become widespread in catheterization laboratories over the next few years.

## 4. Chronic Inflammation, Endothelial Dysfunction and HF

As mentioned, chronic low-grade inflammation significantly contributes to the development of CMD [35]. Concordantly, endothelium-dependent CMD, assessed by a blunted microvascular response to acetylcholine (ACh), has been associated with increased inflammatory markers such as high-sensitivity C reactive protein (hs-CRP) and soluble urokinase-type plasminogen activator receptor (suPAR) [36]. Inflammatory conditions, particularly in the presence of cardiovascular risk factors, are responsible for increased superoxide production by NAD(P)H oxidase (Nox), and consequent endothelial dysfunction though oxidative stress [37]. In detail, Nox activity is enhanced by inflammatory cytokines such as interleukin-6 (IL-6) and tumor necrosis factor-α (TNF-α), and is able to induce p66 Shc, a pro-apoptotic mitochondrial adapter which, in turn, favors ROS production and further upregulates Nox in a vicious cycle [7,38]. Endothelial dysfunction in the context of ROS vascular accumulation results from reduced NO bioavailability due to its conversion to peroxynitrite radicals and endothelial NO synthetase (eNOS) uncoupling [7]. Other endothelium-derived relaxing factors are involved in coronary microvascular tone regulation, such as H_2_O_2_, which derives from superoxide released by endothelial cells in response to shear stress [39,40]. Superoxide synthesis and its subsequent rapid dismutation to H_2_O_2_ represent a physiological pathway for arteriolar flow-induced vasodilation, as long as the Nox system is not hyperactivated, a condition in which excessive ROS production may precipitate oxidative stress and endothelial dysfunction [39]. Nox enzyme isoform upregulation can result from metabolic disturbances and/or impaired hemodynamic homeostasis, which are associated with traditional cardiovascular risk factors [38]. Furthermore, p66 Shc expression is positively modulated at the epigenetic level in diabetic patients, favoring persistent inflammation and unremitting endothelial dysfunction (the “hyperglycemic memory” phenomenon) [38,41]. Aside from classical cardiovascular risk factors, microvascular and epicardial endothelial dysfunction may also arise from long-term air pollutant exposure, specifically, particulate matter (PM), which has been proven to enhance the pro-inflammatory response leading to systemic oxidative stress [42,43,44]. In this regard, Montone et al. provided evidence that higher exposure to air pollutants, such as 2.5 (PM2.5) and PM10, can drive a higher risk of developing coronary events, including coronary vasomotor disorders in the absence of obstructive CAD, by enhancing systemic and local inflammatory processes [42,43,44].

Moreover, infectious diseases, particularly those enhancing a pro-inflammatory and pro-thrombotic milieu, including coronavirus disease 2019 (COVID-19), recently emerged as further contributors to endothelial dysfunction and CMD [45].

Therefore, cardiovascular risk factors and comorbidities of HF play a key pathogenetic role in determining CMD in this setting, particularly that of HFpEF. Elevated concentrations of inflammation biomarkers such as interleukin-6 (IL-6), TNF-α, pentraxin 3, and soluble suppression of tumorigenicity 2 (sST2) have been described in HFpEF patients, while an increase in inflammatory cells expressing CD3, CD11a, and CD45, together with the vascular cell adhesion molecule VCAM-1, and increased oxidative stress been recognized in HFpEF cardiac tissue samples [28,39]. Interleukin-2 (IL-2) has been recently proposed as a predictor of HFpEF development, playing a hypothesized role in chronic microvascular inflammation, while neutrophilic myeloperoxidase (MPO)-related oxidative stress has been associated with endothelial dysfunction in HFpEF [46,47]. In the setting of oxidative stress at the level of the coronary microvasculature, an ROS-mediated reduction in the endothelial NO bioavailability leads to decreased cyclic guanosine monophosphate (cGMP) levels and protein kinase G (PKG) activity, which favors structural cardiac changes such as left ventricular concentric remodeling and hypertrophy [24]. Inflammation-driven PKG activity, such a deficiency also impairs cardiomyocyte relaxation due to titin hypophosphorylation [24]. Simultaneously, impaired endothelial NO release gives rise to fibroblast and myofibroblast proliferation, while NO-mediated antifibrotic effects through the cGMP pathway are blunted [2]. Moreover, the inflammation of the microvascular endothelium itself causes the migration of monocytes and secretion of transforming growth factor-β (TGF-β), favoring interstitial fibrosis [3]. As previously mentioned, CMD induces diastolic dysfunction not only through fibrosis but also through structural microvascular rarefaction, and both processes may arise from a common mechanism involving a microvascular endothelial inflammatory state [5]. Pro-oxidative conditions at the level of diseased coronary microvessels make them prone to microinfarcts during local ischemic events, further enhancing fibrosis and consequent reduced myocardial diastolic function and systolic reserve [5]. Myofibroblasts and inflammatory cells migrating through the endothelium into cardiac interstitial tissue may then fuel inflammation, stimulate fibrosis via TGF-β, and downregulate matrix metalloproteinase (MMP-1), thus reducing extracellular matrix degradation [48]. Consequently, the reciprocal biological signaling taking place between the endothelium, perivascular tissue, and myocardium is decisive in HFpEF pathogenesis and progression.

The role of comorbidity-driven systemic inflammation in the pathophysiology of HFrEF may be less significant than in HFpEF [48,49]. Circulating markers such as IL-6, TNF-α, and CRP are known to be better predictors of HFpEF than of HFrEF, and their levels are related to the overall comorbidity burden of HFpEF patients [50]. On the other hand, oxidative stress directly affects cardiomyocytes in HFrEF, causing apoptosis or necrosis and replacement fibrosis, a feature that is not observed in HFpEF [48]. Of importance, an elevation in inflammatory biomarkers such as TNF-α, soluble TNF-receptor II, and IL-6 has been observed in the advanced stages of HFrEF, and endothelium-dependent vasodilatation impairment has also been demonstrated in HFrEF patients, probably related to oxidative stress [51,52]. Thus, inflammation-related endothelial dysfunction may also be a marker of disease progression and adverse prognosis in patients with HFrEF.

## 5. Targeting CMD in Patients with HF

To date, there are no specific pharmacologic strategies to treat CMD in patients with HF and randomized clinical trials addressing this field are lacking. An accurate understanding of the pathophysiological mechanisms underlying this condition is essential to provide comprehensive and effective therapeutic interventions (both pharmacological and addressing the lifestyle) aimed at targeting different pathophysiological aspects of the disease (Figure 2). 

As mentioned above, patients affected by CMD and HFpEF typically suffer from numerous comorbidities and risk factors. Lifestyle modifications including regular exercise, weight management, and smoking cessation are fundamental to improving endothelial function and have been shown to ameliorate the coronary flow reserve, with beneficial effects on exercise capacity and cardiovascular outcomes [3,51,52,53,54,55].

Nevertheless, risk factors and comorbidities contribute to the creation and perpetration of a pro-inflammatory status, which plays a key role in the development of CMD. Consequently, inflammation represents an important target in CMD management and can be approached with different drugs. In patients with high blood cholesterol, statins have been shown to exert anti-inflammatory properties and positive effects on endothelial homeostasis by restoring the downstream NO signaling and improving endothelium-dependent coronary relaxation [3,6]. A meta-analysis and observational studies recently demonstrated that statin treatment may reduce mortality and HF hospitalizations among patients with HFpEF, with a beneficial effect that seems independent from cholesterol levels and HF severity and likely due to their off-target anti-inflammatory properties [56,57,58]. In an interesting study by Eshtehardi et al., the imaging characteristics of coronary atheromas were analyzed at baseline and after 6 months of high-intensity statin treatment (atorvastatin 80 mg) among 20 patients with moderate CAD [57]. The authors proved that patients on statin treatment showed not only a modification in atheroma composition and plaque phenotype but also a modest improvement in coronary microvascular function, assessed by coronary flow reserve (+0.26 [IQR, −0.37 to 0.76]; *p* = 0.23) and hyperemic microvascular resistance (−0.22 [IQR, −0.56 to 0.28]; *p* = 0.12) [57].

Beta-blockers are a cornerstone of the pharmacological management of HFrEF, while there are no specific data on their potential beneficial effects in HFpEF. In patients with microvascular angina (MVA), beta-blockers, in particular third-generation beta-blockers such as nebivolol and carvedilol, have been shown to ameliorate effort-related symptoms and exercise tolerance by improving endothelial function, decreasing myocardial oxygen needs and extending diastolic perfusion time [59,60,61]. Therefore, they are recommended as first-line therapy in this group of individuals [61]. However, beta-blockers should not be used routinely in HFpEF since they ultimately reduce cyclic AMP, which is important for myocardial relaxation and diastolic function [62,63].

Angiotensin-converting enzyme inhibitors (ACEis) are strongly recommended as the first-line therapy for patients with HFrEF, given their demonstrated efficacy in reducing cardiovascular mortality and hospitalization for HF on a large-scale basis [2,64]. Data from the Trial on Reversing Endothelial Dysfunction (TREND) study further support the positive impact of ACE inhibitors, particularly quinapril, on endothelial function. This study, involving 101 patients without HF, hypertension, or dyslipidemia, underscored the beneficial effects of ACE inhibitors on endothelial function by evaluating the effect of quinapril (40 mg daily) vs. placebo on coronary artery responses to acetylcholine using quantitative coronary angiography. The authors proved that after 6 months of therapy, only the quinapril group showed a significant enhancement in the vasoactive response to incremental concentrations of acetylcholine (*p* = 0.002) [64]. These beneficial properties of ACE inhibitors on CMD were attributed to their ability to inhibit coronary vasoconstriction and superoxide generation while enhancing endothelial cell release of NO and further improving endothelial homeostasis.

Sacubitril/valsartan is a first-class recommendation in patients with HFrEF as it has been shown to reduce all-cause mortality and HF hospitalization compared to enalapril [2,65]. In patients with HFpEF, the PARAMOUNT-HF Phase II trial has demonstrated a significant reduction in NT-proBNP levels after 12 weeks of therapy with sacubitril-valsartan in comparison with valsartan (baseline: 783 pg/mL [95% CI 670–914], 12 weeks: 605 pg/mL [95% CI 512–714] vs. baseline: 862 pg/mL [95% CI 733–1012], 12 weeks: 835 pg/mL [95% CI 710–981], respectively, *p* = 0.005) [66]. Conversely, in the PARAGON-HF trial, which included 4822 patients with HF and an ejection fraction of 45% or higher, sacubitril/valsartan failed to meet the primary endpoint of a reduction in the total hospitalizations for HF (690 vs. 797 total hospitalizations for heart failure, respectively [rate ratio, 0.85; 95% CI, 0.72 to 1.00]) and death from cardiovascular causes (8.5% in the sacubitril–valsartan group vs. 8.9% in the valsartan group [hazard ratio, 0.95; 95% CI, 0.79 to 1.16]) compared to valsartan [67]. Notably, the impact of angiotensin receptor-neprilysin inhibitor (ARNI) therapy on the emerging pathophysiological mechanisms underlying HFpEF, such as myocardial interstitial fibrosis and CMD, remains largely unexplored. Addressing this gap, ongoing research like the PRISTINE-HF Study (Study of Sacubitril/ValsarTan on MyocardIal OxygenatioN and Fibrosis in Heart Failure with Preserved Ejection Fraction) will evaluate the potential effects of sacubitril-valsartan on CMD and microvascular ischemia using stress perfusion oxygen-sensitive CMR in HFpEF patients (NCT04128891).

Recently, sodium glucose co-transport-2 inhibitors (SGLTis), a class of antidiabetic drugs, have been introduced as first-line therapy in patients with HFrEF as they have been shown to reduce the all-cause mortality and HF hospitalization rate in this group of patients [3,68,69,70]. Preclinical studies utilizing a pre-diabetic murine model have also suggested potential positive effects of SGLT2is on coronary microvascular function, revealing a significant improvement in the coronary flow reserve in the animals treated [71]. More recently, the Effect of Empagliflozin on Worsening Heart Failure Events in Patients With Heart Failure and Preserved Ejection Fraction trial (EMPEROR-Preserved trial) and the Efficacy and Safety of Dapagliflozin in Heart Failure With Mildly Reduced or Preserved Ejection Fraction According to Age trial (The DELIVER Trial) have corroborated these findings, reporting significant reductions in cardiovascular mortality and the rates of heart failure hospitalization in HFpEF patients receiving SGLT2i therapy [72,73]. Consequently, the most recent 2023 updated guidelines on heart failure management from the ESC integrated this class of drugs as a first-line therapy for HFpEF [74]. The pleiotropic effects of SGLT2is are still largely unknown. Nevertheless, emerging data suggest that the inhibition of pro-inflammatory cytokines and ROS production can contribute to off-target anti-inflammatory properties having additional beneficial effects on endothelial homeostasis [75,76]. Indeed, in animal models, SGLT2is have been proven to reduce the circulating levels of pro-inflammatory biomarkers, such as IL-6 TNF and C-reactive protein, with well-known detrimental effects on endothelial function [75]. 

In addition to SGLT2is, another class of antidiabetic drugs, Glucagon-Like Peptide Receptor Agonists (GLP-1 RAs), seem to have promising cardiovascular effects. Encouraging data on cardiovascular outcomes have emerged in several trials on diabetic patients [77,78,79,80]. The pathophysiological rationale is to be found in the endothelial expression of GLP-1 RAs. They are able to decrease ROS production, leading to reduced oxidative stress, with a positive effect on the atherosclerotic process [81]. GLP-1 RAs also modulate inflammation with a direct action on inflammatory cells [82,83]. Furthermore, the use of Liraglutide seems to improve diastolic function [84,85,86], typically impaired in HFpEF patients. However, to date, randomized studies with GLP-1 RA in HFpEF patients are still lacking.

Several other anti-anginal drugs, such as ivabradine, ranolazine, nicorandil, and trimetazidine, have been investigated in patients with CMD with spare data to date in the setting of HF pathology [3,87]. Ivabradine is a selective inhibitor of the hyperpolarization-activated cyclic-nucleotide gated funny current (If) in the sinoatrial node, with the known property of heart rate reduction. According to the ESC guidelines on HF management, it is indicated with a class II level of recommendation A in patients with HFrEF when beta-blockers are contraindicated or as additional anti-anginal therapy in patients with sinus rhythm to reduce the heart rate and oxygen consumption [2]. However, ivabradine has also been shown to have beneficial effects on MVA [3,88]. In a randomized controlled trial evaluating the effects of ranolazine and ivabradine vs. placebo among 46 patients with stable MVA (effort angina, positive exercise stress test [EST], normal coronary angiography, coronary flow reserve < 2.5) and anginal symptoms despite standard anti-ischemic medical therapy, ivabradine was proven to effectively reduce angina symptoms and quality of life assessed by the Seattle Angina Questionnaire (SAQ) and EuroQoL scale (*p* < 0.05) [88]. According to the ESC guidelines on HF management, it is indicated with a class II level of recommendation A in patients with HFrEF when beta-blockers are contraindicated or as additional anti-anginal therapy in patients with sinus rhythm [2]. Ranolazine has been tested in several small randomized controlled trials, with controversial results in terms of coronary flow reserve and symptom improvement [3,89]. Nitrates have demonstrated inconclusive results in CMD but their use in the short-acting form can be considered to treat angina attacks, especially among patients with an impaired vasodilator reserve [3]. Fasudil is an inhibitor of the Rho-kinase pathway, a molecular pathway with a pivotal role in the pathogenesis and development of MVA that modulates the calcium sensitivity of the myosin light chain in smooth muscle cells [3]. In a study by Mohri et al. evaluating the effect of the Rho-kinase inhibitor in 18 patients with angiographically proven MVA, the pretreatment with Fasudil was proven to be highly effective in preventing acetylcholine-induced coronary vasospasm compared to placebo (saline solution) (*p* < 0.01) and it can be considered in patients with MVA [90]. These results were further strengthened by other studies with a larger sample size, corroborating the promising therapeutic role of this class of drugs in patients with CMD on top of the standard anti-ischemic medical therapy [91]. 

## 6. Conclusions and Future Perspectives

In conclusion, CMD is an emerging clinical condition due to structural and/or function alterations at the coronary microcirculatory level which represents a significant and often-overlooked component of HF pathophysiology, contributing to myocardial ischemia and fibrosis, impaired cardiac function, and adverse clinical outcomes. The precise mechanisms linking CMD to heart failure development and progression are still not well known and randomized clinical trials evaluating the efficacy of specific treatments targeting these molecular mechanisms are lacking to date. However, comorbidities promoting systemic and coronary inflammation (such as diabetes, chronic kidney disease, and infections) seem to play a pivotal role as predisposing factors in the pathogenesis of endothelial dysfunction leading to CMD and finally HF, especially HFpEF. This is likely the HFpEF phenotype in which a systematic assessment may be worthwhile. Ongoing clinical trials assessing the efficacy of emerging pharmacological therapies for HF, such as SGLT2is and ARNIs, in preserving microvascular integrity and function hold promise for advancing our understanding and management of CMD in this particular clinical setting. Overall, continued efforts to unravel the complexities of CMD in HF are crucial for improving patient outcomes and reducing the burden of this debilitating condition.

## Figures and Tables

**Figure 1 ijms-25-07628-f001:**
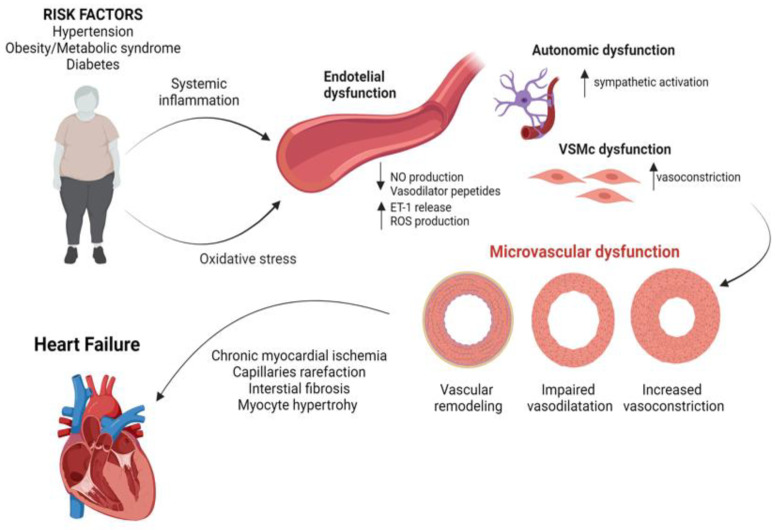
This figure illustrates the pathophysiological mechanisms linking heart failure and CMD. Created with Biorender.com. Abbreviations: CMD: coronary microvascular dysfunction.

**Figure 2 ijms-25-07628-f002:**
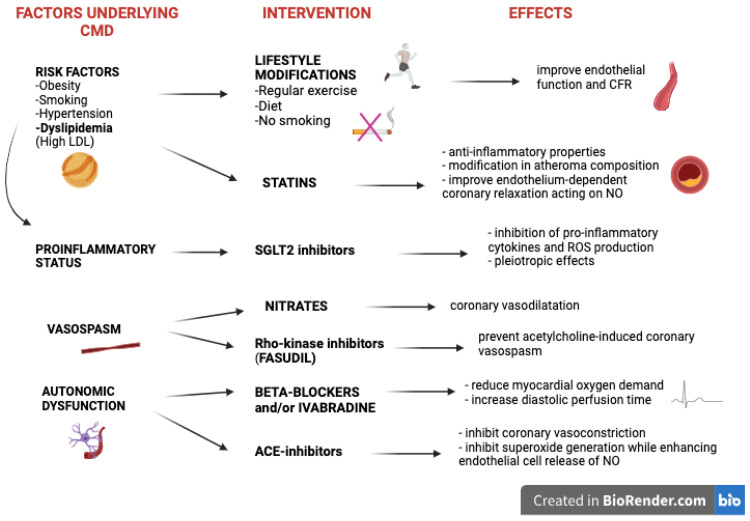
This figure illustrates the pathophysiological mechanisms underlying CMD and the related interventions (pharmacological or otherwise), with the expected effects. Abbreviations: ACE: Angiotensin-converting enzyme; CFR: coronary flow reserve; CMD: coronary microvascular dysfunction; NO: nitric oxide; SGLT-2: Sodium-Glucose Transport Protein 2.

**Table 1 ijms-25-07628-t001:** Studies evaluating the prevalence and prognostic relevance of CMD in HFpEF.

Authors	Year	Sample Size (n)	Study Design	Age (Years)	Objective	Definition of CMD	Prevalence of CMD	Results	Reference
Rush et al.	2021	106	nROS	72 ± 9	To assess the prevalence of CAD and CMD in hospitalized patients with HFpEF	CMD defined both invasively (as CFR < 2.0 and/or iMR > 25 and/or positive Ach provocative test) and non-invasively (as MPRi < 1.84 at CMR perfusion imaging)	85%	CMD is highly prevalent in HFpEF with and without CAD	[4]
Mohammed et al.	2015	228	nROS	75 (66–83)	To evaluate structural changes associated with HFpEF	Microvascular rarefaction defined as vessels/mm^2^ at myocardial biopsy	23%	HFpEF showed a high prevalence of myocardial fibrosis and coronary microvascular rarefaction	[5]
Shah et al.	2018	202	nROS	74.7 ± 8.7 in CMD pts vs. 72.4 ± 9 in those without	To investigate the prevalence of CMD and its association with endothelial dysfunction and HF severity in HFpEF	CMD defined as CFR < 2.5 assessed by stress transthoracic echocardiography	75%	CMD is highly prevalent in HFpEF and is associated with clinical biomarkers of HF severity	[24]
Lin et al.	2023	1267	Systematic review and meta-analysis	N/A	To assess the prevalence of CMD in HFpEF	CMD assessed both invasively (including both cut-offs of CFR < 2.0 and CFR < 2.5) and non-invasively	71%	CMD is highly prevalent in HFpEF and is associated with worse clinical outcomes	[25]
Yang et al.	2020	162	nROS	56 ± 11 in CMD pts vs. 54 ± 11 in those without	To assess the prevalence of endothelium-dependent vs. -independent CMD in HFpEF	CMD defined invasively by Ach provocative test	72%	Endothelium-dependent and -independent CMD are equally prevalent in HFpEF	[26]
Arnold et al.	2021	144	nROS	73 ± 5	To examine the prevalence of CMD, the relationship between perfusion and fibrosis, and the impact of CMD on clinical outcomes in HFpEF	CMD defined as MPR < 2.0 assessed by stress perfusion CMR study	70%	CMD is highly prevalent in HFpEF (up to 70% of cases) and is independently associated with worse clinical outcomes	[27]
Paolisso et al.	2024	56	nROS	N/A	To characterize coronary CMD in HFpEF vs. HFrEF	CMD defined as CFR < 2.5 assessed invasively by intracoronary thermodilution	52%	In HFrEF, CMD was mainly functional while in HFpEF, it was mainly characterized by structural changes	[28]
Srivaratharajah et al.	2016	376	nROS	63 ± 11	To assess myocardial flow reserve (MFR) in HFpEF	MFR > 2.0 assessed by cardiac positron emission tomography	40%	HFpEF was associated with a significant reduction in global MFR	[29]
Dryer et al.	2018	44	nROS	65.4 ± 9.6 in HF pts vs. 55.1 ± 3.1 in controls	To assess the prevalence of CMD in HFpEF	CMD defined invasively as CFR < 2.0 and/or iMR > 23	37% with overt CMD and 37% with either abnormal iMR or CFR	Distinctive coronary physiology groups are present in HFpE	[30]
Kato et al.	2021	163	nROS	73 ± 9	To assess the prognostic value of CMD in HFpEF	CMD defined as CFR < 2 assessed by stress perfusion CMR	9%	CFR is a valuable prognostic marker in HFpEF	[31]
Ahmad et al.	2021	51	nROS	59.6 ± 10.1 in pts with diagnosis of HFpEF vs. 54.3 ± 10.4 in pts without	To assess the relationship between microvascular function and exercise hemodynamics	CMD defined invasively as CFR ≤ 2.5 and/or abnormal Ach provocative test	86%	CMD is associated with higher left ventricular filling pressures at peak exercise level	[32]
Mohammed et al.	2023	137	nROS	N/A	To assess the prognostic significance of CMD in HFpEF	CMD defined as coronary angiography-derived index of microcirculatory resistance ≥ 25	64%	CMD is an independent prognostic predictor of HFpEF	[33]

Abbreviations: CAD: coronary artery disease; CMR: Cardiovascular Magnetic Resonance; CFR: coronary flow reserve; HFpEF: heart failure with preserved ejection fraction; HFrEF: heart failure with reduced ejection fraction; iMR: index of microvascular resistance; MPR: myocardial perfusion reserve; MPRi: myocardial perfusion reserve index; CMD: microvascular disease.

## Data Availability

No new data were created.

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
