# Peer review of "Microvascular Dysfunction across the Spectrum of Heart Failure Pathology: Pathophysiology, Clinical Features and Therapeutic Implications"

_ijms, 2024, doi:10.3390/ijms25147628_

Round 1

Reviewer 1 Report

Comments and Suggestions for Authors

Major comment:

A nice manuscript but lacking some key references in the area. Also novel treatment targets may be broaden to ongoing programs.

Specific points:

Background:

Would recommend including reference DOI: 10.1002/ehf2.14626 at line 37 and reference DOI: 10.1016/j.jacc.2013.02.092 or DOI: 10.1161/circresaha.121.318159 at line 39, the latter key references in the area.

Key/original references on

Endothelium-dependent and independent coronary microvascular dysfunction DOI:10.1002/ ejhf.1671

The role of endothelial activation and oxidative stress and of NO-dependent signalling DOI:10.1016/j.jchf.2015.10.007, lines 68-69

Would object to the statement “More recently…” at line 120. Paulus key reference is more than a decade. An updated presentation of the paradigm was published 2021 DOI:10.1161/circresaha.121.318159

At line 125; Demonstrated that inflammation mediate the association between comorbidity burden and cardiac dysfunction DOI: 10.1161/circulationaha.120.045810

Lines 289-295 – Betablockers in HFpEF have been questioned – which should be discussed. Unless indicated for specific comorbidities there are data supporting non-use of betablocker in HFpEF for symptomatic improvement and increased exercise capacity doi.org/10.1007/s00392-024-02396-4

Regarding novel treatments – Inflammation as a treatment target may be discussed doi: 10.1016/j.ijcard.2024.131834.

Table 1

(repeated in the Table heading – which is below the Table?) presents prevalence and prognostic relevance of CMD in HFpEF.

Unclear how the presented studies are selected as there are several more in the same area, such as:

Srivaratharajah 2016 doi:10.1161/CIRCHEARTFAILURE.115. 002562

Dryer 2018 doi:10.1152/ajpheart.00680.2017

Allan 2019 doi:10.1016/j.cardfail.2019. 08.010

Yang 2019 doi:10.1002/ ejhf.1671

Mahfouz 2020 doi:10.1111/echo.14799

Kato 2021 doi:10.1186/ s12968-021-00807-3

Ahmad 2021 doi:10.1002/ejhf.2010

Mohammed, 2023 doi:10.1016/ j.cjca.2023.04.011

Rezaeian 2024 doi: 10.1016/j.ahjo.2024.100390. eCollection 2024 May.

Author Response

Would recommend including reference DOI: 10.1002/ehf2.14626 at line 37 and reference DOI: 10.1016/j.jacc.2013.02.092 or DOI: 10.1161/circresaha.121.318159 at line 39, the latter key references in the area.

R: We thank the review for his/her helpful comment to improve our manuscript. After careful revision, we decided to cite the suggested references except for DOI:10.1016/j.jacc.2013.02.092 as too dated.

 Key/original references on Endothelium-dependent and independent coronary microvascular dysfunction DOI:10.1002/ ejhf.1671

The role of endothelial activation and oxidative stress and of NO-dependent signalling DOI:10.1016/j.jchf.2015.10.007, lines 68-69

R: We thank the review for his/her suggestion. After careful revision, we decided to cite the reference DOI:10.1002/ ejhf.1671 and not to cite DOI:10.1016/j.jchf.2015.10.007 as too dated.

Would object to the statement “More recently…” at line 120. Paulus key reference is more than a decade. An updated presentation of the paradigm was published 2021 DOI:10.1161/circresaha.121.318159

R: We do appreciate the review suggestion. Accordingly, we replace the publication by Paulus et al. with DOI:10.1161/circresaha.121.318159.

Lines 289-295 – Betablockers in HFpEF have been questioned – which should be discussed. Unless indicated for specific comorbidities there are data supporting non-use of betablocker in HFpEF for symptomatic improvement and increased exercise capacity doi.org/10.1007/s00392-024-02396-4

R: We thank the review for his/her suggestion. We have updated the relevant part.

Regarding novel treatments – Inflammation as a treatment target may be discussed doi: 10.1016/j.ijcard.2024.131834.

R: We thank the review for his/her suggestion. We have updated the relevant part.

Table 1

(repeated in the Table heading – which is below the Table?) presents prevalence and prognostic relevance of CMD in HFpEF.

Unclear how the presented studies are selected as there are several more in the same area, such as:

Srivaratharajah 2016 doi:10.1161/CIRCHEARTFAILURE.115. 002562

Dryer 2018 doi:10.1152/ajpheart.00680.2017

Allan 2019 doi:10.1016/j.cardfail.2019. 08.010

Yang 2019 doi:10.1002/ ejhf.1671

Mahfouz 2020 doi:10.1111/echo.14799

Kato 2021 doi:10.1186/ s12968-021-00807-3

Ahmad 2021 doi:10.1002/ejhf.2010

Mohammed, 2023 doi:10.1016/ j.cjca.2023.04.011

Rezaeian 2024 doi: 10.1016/j.ahjo.2024.100390. eCollection 2024 May.

R: We thank you the reviewer for his/her valuable comment. We included some of the most relevant proposed studies in Table 1.

Reviewer 2 Report

Comments and Suggestions for Authors

This review article has aims to explore the molecular mechanisms underlying coronary microvascular dysfunction (CMD), and its distinct manifestations across the spectrum of heart failure (HF) pathology, discussing novel therapeutic targets and remaining knowledge gaps in the field.

This reviewer considers that this review article was well written and has a comment as described below:

Major comment:

1.       Although the authors mentioned that the invasive and non-invasive diagnosis of CMD is still challenging, but as a review article, they should add the section regarding diagnosis of CMD in this article.   

Author Response

We thank the reviewer for his/her valuable comment. We appreciate his concerns about the lack of information regarding the diagnostic work-up of CMD in HFpEF patients as it is of utmost importance in the clinical scenario discussed in this paper. However, due to the complexity of the topic, we decided to focus on pathophysiology, clinical features and therapeutic implications. We improved the review’s title accordingly to make it clearer

Reviewer 3 Report

Comments and Suggestions for Authors

The Authors submitted a review on microvascular dysfunction across the spectrum of heart failure  pathology. The manuscript is well written and represents an useful update for readers.

A few comments

1) Table 1. I suggest to add details on study results (i.e., %, HRs etc) to give more context to readers.

2) The Authors should discuss more on the strength and limitation of longitudinal strain in CMD and HFpEF referring also to Sanna GD et al. Curr Heart Fail Rep. 2021 Oct;18(5):290-303.

3) What about the potential role of GLP-1RAs for CMD in HFpEF patients? 

4) Finally, what's the HFpEF phenotype where the systematic CMD assessment may be worthwhile?

Comments on the Quality of English Language

Minor editing of English language required

Author Response

1)Table 1. I suggest to add details on study results (i.e., %, HRs etc) to give more context to readers.

R: We thank the reviewer for his/her comment. We provided more information in Table 1 to make it more informative (e.g. design of study, mean age, prevalence of CMD). We decided to not provide HR values as not all studies included were prognostic.

2) The Authors should discuss more on the strength and limitation of longitudinal strain in CMD and HFpEF referring also to Sanna GD et al. Curr Heart Fail Rep. 2021 Oct;18(5):290-303.

R: We thank the reviewer for his/her valuable comment. Despite the interesting data provided by the suggested paper, we decided not to discuss it as our review focuses on pathophysiology, clinical features and therapeutic implications of CMD in HFpEF and not on diagnostic work-up of the latter (including novel echocardiographic indices as GLS).

3) What about the potential role of GLP-1RAs for CMD in HFpEF patients?

R: We thank the reviewer for his/her valuable comment. We provided information about potentiality of this class of drugs.

4) Finally, what's the HFpEF phenotype where the systematic CMD assessment may be worthwhile?

R: We thank the reviewer for his/her valuable comment. We have underlined the correlation between CMD and HFpEF phenotype characterized by multiple comorbidities

Round 2

Reviewer 1 Report

Comments and Suggestions for Authors

The manuscript has been nicely revised and updated with relevant references. Still one pending question from this reviewer:

Unclear how the presented studies were selected – were there a specific set of criteria?

Author Response

We thank the reviewer for his/her valuable comment.

The studies presented in our work were selected based on a predefined set of criteria designed to ensure relevance and comprehensiveness. Specifically, we employed the following criteria for study inclusion:

1)Relevance to Research Focus: Each study was required to directly address the research questions or objectives outlined in our manuscript (to outline the prevalence and pathophysiological relevance of CMD in HFpEF).

2)Publication Date: Most studies included were published within the last 5 years to ensure the inclusion of the most current research findings.

3) Study Design: Only studies employing scientific rigorous methodologies were considered.

4) Language: We included studies published in English to encompass a broad spectrum of relevant research.

5) Peer-Reviewed: Only studies published in peer-reviewed journals were included to ensure the quality and reliability of the information

Reviewer 2 Report

Comments and Suggestions for Authors

This reviewer has no further comment. 

Author Response

We thank the reviewer for his/her time.